# Hessian-aware training for enhancing model resilience for in-memory computing

## Abstract

Deep neural networks are not resilient to bitwise errors in their parameters: even a single-bit error in their memory representation can lead to significant performance degradation. This susceptibility poses great challenges in deploying models on emerging computing platforms, such as in-memory computing devices, where frequent bitwise errors occur. Most prior work addresses this issue with hardware or system-level approaches, such as additional hardware components for checking a model's integrity at runtime. However, these methods have not been widely deployed since they necessitate substantial infrastructure-wide modifications. In this paper, we study a new approach to address this challenge: we present a novel training method aimed at enhancing a model's inherent resilience to parameter errors. We define a *model-sensitivity* metric to measure this resilience and propose a training algorithm with an objective of minimizing the sensitivity. Models trained with our method demonstrate increased resilience to bitwise errors in parameters, particularly with a 50% reduction in the number of bits in the model parameter space whose flipping leads to a 90–100% accuracy drop. Our method also aids in extreme model compression, such as lower bit-width quantization or pruning ∼70% of parameters, with reduced performance loss. Moreover, our method is compatible with existing strategies to mitigate this susceptibility.

## 1 Introduction

Recent studies have shown that deep neural networks (DNNs) are *not* resilient to bitwise errors in their parameters. For example, a single-bit error on the memory representation of a model parameters can significantly reduce a DNN's performance at inference (Hong et al., 2019; Rakin et al., 2019). This lack of resilience poses a challenge for deploying DNN models to emerging computing platforms, such as in-memory computing devices or neuromorphic computing platforms (Sebastian et al., 2020; Xu et al., 2021; Aguirre et al., 2024), where bitwise errors in devices are frequent (Yao et al., 2020a).

Most prior work addresses this reliability issue by developing defensive mechanisms at the hardware-level or system-level (Bennett et al., 2021; Rakin et al., 2021; Li et al., 2021; Di Dio et al., 2023; Zhou et al., 2023a;b; Liu et al., 2023; Wang et al., 2023). While having demonstrated their effectiveness, these approaches are often difficult to implement in practice as they require additional hardware components or updates to system software, necessitating infrastructure-wide changes.

In this work, we study an *orthogonal* approach to address this resilience problem. We ask: *How can we train models to have increased resilience to bitwise errors in parameters?* No prior work has studied solutions to enhance the natural resilience of a model to bitwise errors in its parameters. Such resilient models will also benefit model compression techniques, e.g., quantization, which involve optimal parameter variations (LeCun et al., 1989). Moreover, when combined with existing solutions, it will complement them and enable models to maintain stable performance in error-prone hardware.

**Contributions.** *First*, we present a novel training algorithm that enhances a DNN's resilience to bitwise errors in their parameters. We focus on a model's second-order property—the *sharpness*—that can approximate the sensitivity of a model's performance to its parameter variations. We empirically test various approaches to reducing the sharpness, from training with second-order optimizers (Yao et al., 2021a) to training algorithms specifically designed to reduce the value (Foret et al., 2021).

Our extensive analysis shows that using the Hessian trace as a loss function is the most effective in decreasing a model's sharpness. However, computing the *full* Hessian matrix makes the training

process computationally intractable, especially when training ImageNet-scale models. To address this challenge, we also develop an optimized training strategy that minimizes only the largest-$p\%$ of the Hessian eigenvalues during training (for example, $p$ is 10–50% in our evaluation).

*Second*, we comprehensively evaluate the effectiveness of our approach across multiple datasets and network architectures including ones used in prior studies. We adapt the systematic resilience analysis framework developed by Hong et al. (2019): the framework examines a single-bit error—an atomic error that can occur in a DNN's parameter representation in memory. It causes all possible single-bit errors one-by-one and measure the accuracy on a validation set each time.

We demonstrate that our training algorithm significantly enhances a model's resilience to (individual) single-bit errors to its parameters. The number of parameters whose perturbations can cause the accuracy drop over 10% are reduced by 5–15%. Particularly, we reduce by half the number of parameters when a single-bit error in them causes the accuracy drop of 90–100%. ImageNet models, fine-tuned only a few layers with our Hessian loss, show a similar decrease in such parameters.

We conduct an in-depth analysis of the increased resilience achieved by our approach. In our analysis of visualized loss landscapes, we show that the sharpness is greatly decreased across all the layers in a model. We find that our approach is particularly effective on fully-connected layers and the convolutional layers in architectural blocks without skip connections. Moreover, we show that the numerical changes in parameter values required to cause a significant accuracy drop has increased.

*Third*, we also demonstrate that the resilience enhanced by our approach can benefit the techniques that rely on *optimal brain damage*, such as quantization (Fiesler et al., 1990) or pruning (Han et al., 2015). Models trained with our algorithm achieve better test accuracy than regularly-trained models, especially when lower bit-width quantization (e.g., 4- or 2-bit) is applied. In pruning, these models preserve test accuracy even with extreme sparsity values (i.e., removing ∼70% of parameters).

*Fourth*, we discuss the compatibility of our approach with existing defensive mechanisms for improving DNN resilience to bitwise errors in model parameters. For each of the hardware-level or system-level mechanisms, we conceptually demonstrate how the models trained with our approach can be combined with them and offer a synergy while reducing their deployment overheads.

## 2 RELATED WORK

LeCun et al. (1989) demonstrates the resilience of DNNs to *optimal brain damage*: one can remove a large portion of the model parameters without causing any significant accuracy drop. This property has enabled the success of model compression techniques, such as quantization (Fiesler et al., 1990; Morgan et al., 1991; Courbariaux et al., 2015) and pruning (Hassibi and Stork, 1992; Han et al., 2015; Li et al., 2017). Prior work further leverages this property to safeguard DNNs against potential threats, such as adversarial examples (Zhou et al., 2018) while preserving the performance.

Recent work has also warned of the possibility of *terminal brain damage* (Hong et al., 2019; Yao et al., 2020b), where even a few perturbations to parameters can cause significant performance loss. It is also important to note that the prevalence of such failures: half of model parameters, when subject to a bitwise corruption, cause the accuracy to drop over 10% (Hong et al., 2019). Beyond its impact on model accuracy, follow-up works have shown that by perturbing parameters in specific ways, one can achieve various adversarial outcomes (Rakin et al., 2019; Chen et al., 2021; Rakin et al., 2022).

There has been significant effort in mitigating the impact of adversarial perturbations on model parameters through hardware (Kim et al., 2014; Bennett et al., 2021; Saileshwar et al., 2022; Di Dio et al., 2023; Zhou et al., 2023b) and software (Liu et al., 2023; Li et al., 2021; Konoth et al., 2018) level defenses. Hardware-level defenses utilize proactive row refreshing (Kim et al., 2014), in-DRAM counter-based mitigation (Bennett et al., 2021), risky-row swapping (Saileshwar et al., 2022), error-correction-codes (ECC) for swap triggering (Di Dio et al., 2023) and utilizing lock table for high-risk rows inside a memory device (Zhou et al., 2023b) to mitigate adversarial bit-flipping on model parameters. System-level defenses use techniques, such as luring adversary to manipulate "honeypot" parameters (Liu et al., 2023), checksum-based runtime checks to detect changes in model parameters (Li et al., 2021) and employing data-row isolation to safeguard a model's parameters from fault attacks (Konoth et al., 2018). A few works explore techniques for improving thefault tolerance of DNNs. Buldu et al. (2022) adapts adversarial training to train models under bitwise errors. Chitsaz et al. (2023) proposes learnable quantization to limit the impact of bitwise errors on DNN inference.

Our work does *not* focus on defending DNNs against adversarial bit-flip attacks. Instead, we are concerned with unintended bitwise errors that frequently occur in emerging computing platforms.

# 3 EXPERIMENTAL SETUP

**Datasets.** We use three datasets designed for benchmarking image classification: MNIST (LeCun et al., 2010), CIFAR-10 (Krizhevsky, 2009), and ImageNet (Russakovsky et al., 2015).

**Models.** We experiment with four different DNNs typically used in the prior work on evaluating the resilience to parameter corruptions. For MNIST, we employ two feed-forward DNNs: one with two convolutional layers and two fully-connected layers, and LeNet (Lecun et al., 1998). For CIFAR-10 and ImageNet, we consider a DNN architecture popular in the community, ResNets (He et al., 2016). We also run our evaluation on a Transformer-based model: DeiT-Tiny (Touvron et al., 2021).

**Metrics.** We introduce the evaluation metrics here to establish a clear framework for assessing our approach's effectiveness before discussing our methodology. We follow the prior work (Hong et al., 2019) to quantify the resilience of a model under bitwise errors in its parameters. To quantify performance degradation, we measure the relative accuracy drop (RAD) that computes $(A_c - A_p)/A_c$, where $A_c$ is the classification accuracy of a model on a validation set and $A_p$ is the accuracy of the model under parameter corruptions. We also define a *erratic parameter* as the parameter under a single-bit error can lead to RAD over 10%. Because most prior work considers a 10% RAD significant, we use this 10% threshold to determine the resilience.

We also define *the distribution plot* where we count the number of bits in a model's memory representation whose flipping leads to RAD specified in the x-axis. Figure 1 shows the distribution plots contrasting the two MNIST models, one trained with our Hessian-aware training method and the other not. We use a 5% granularity on the x-axis for our plots. The plot shows that our training method can overall reduce the total number of bits whose errors result in a relative accuracy drop and also decrease the number of bits whose flipping leads to a 95–100% accuracy drop. By using this plot, we gain a deeper understanding of the severity and impact of parameter perturbations before and after the application of our training algorithm.

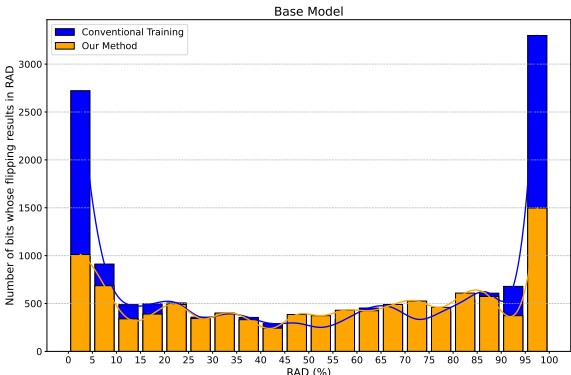

Figure 1: **The distribution plots.** Each plot shows the number of bits in a DNN's parameters whose flipping results in RAD specified in the x-axis.

# 4 OUR APPROACH: HESSIAN-AWARE TRAINING

## 4.1 A DNN'S SENSITIVITY TO PARAMETER VARIATIONS

We first focus on functions that can quantify a model's sensitivity to parameter value variations. Our goal here is to use these functions as optimization objectives to minimize sensitivity during training. Suppose that a model $f$ uses a loss function $\mathcal{L}$. The rate of change in the loss in a random direction $v$ in the parameter space can be expressed as the gradient $\partial \mathcal{L}/\partial v$. This value encodes how sensitive a model will be when its parameter values are changed along the direction of $v$. During optimization, the training algorithm seeks a minimum that reduces this rate of change. However, standard training does not inherently offer the resilience of a model to parameter variations caused by bitwise errors.

**Second-order derivative as a sensitivity metric.** The loss curvature of a neural nework is typically super-linear, forming a convex hull around the local minima in the parameter space (Li et al., 2018). We thus use the second-order derivative of the loss function in a random direction $v$ as a metric to quantify a model's sensitivity to parameter variations. We note that this property has been used in prior work (Jiang et al., 2020; Mulayoff and Michaeli, 2020; Li et al., 2018; Keskar et al., 2017; Neyshabur et al., 2017) as a measure of the *sharpness* or *flatness* of the loss landscape. Few prior

works (Foret et al., 2021; Dong et al., 2019; 2020; Yang et al., 2022) have also used second-order property of DNNs to improve generalization, accuracy or model compression. Most prior work studies the interaction between the sharpness and the generalization of a neural network, a few works studies its connection to model resilience to bitwise errors in parameters.

Our work utilizes the Hessian trace, the sum of the eigenvalues of the Hessian matrix. This requires computing the Hessian matrix, the second-order partial derivatives of a loss function, with respect to its parameters. Recent work has also leveraged the Hessian trace to measure the quantizability of a neural network (Dong et al., 2019; 2020; Yao et al., 2021b), and we follow this insight to measure sensitivity. Due to the large number of model parameters (typically ranging from millions to billions), computing the Hessian trace directly is computationally intractable. We use the Hutchinson's method (Bekas et al., 2007) to approximate the Hessian trace over a number of random vectors $v$. Following these prior practices, we use the Hessian trace to quantify sensitivity to parameter value variations.

### 4.2 MINIMIZING A DNN'S SENSITIVITY TO PARAMETER VARIATIONS

We present our novel training algorithm for reducing the sensitivity of a model to its parameter value variations. Our strategy is to employ the Hessian trace as an additional regularization term in the loss function used for training. The training process is shown in Algorithm 1.

---

**Algorithm 1** Hessian-aware Training

---

**Input:** A model $f$, Training data $D$, Training steps $T$, Learning rate $\eta$, Number of approximation steps $N_v$, Regularization coefficient $\lambda$
**Output:** A trained model $f_\theta$

1: Initialize $\theta_0$
2: Initialize $\tau$ to 0
3: **for** $t = 1, 2, ..., T$ **do**
4:     Draw a mini-batch $S_t$ from $D$
5:     Compute the loss $\mathcal{L}_{xe}(S_t; f_{\theta_t})$
6:     $Tr_t, E_t \leftarrow 0, \phi$
7:     **for** $i = 1, 2, ..., N_v$ **do**
8:         Draw a vector $v_i$
9:         Compute the gradient $g_i$ of the loss $\mathcal{L}_{xe}$
10:        Compute the Hessian matrix $H_i$ along $v_i$
11:       Compute their eigenvalues $E_i$ and trace $Tr_i$
12:       $Tr_t, E_t \leftarrow Tr_t + Tr_i, E_t + E_i$
13:     **end for**
14:     $Tr_t, E_t \leftarrow (1/N_v)Tr_t, (1/N_v)E_t$
15:     **if** $\text{Median}(E_t) > \tau$ **then**
16:        $\mathcal{L}_{tot} \leftarrow \mathcal{L}_{xe}(S_t; f_{\theta_t}) + \lambda * Tr_t$
17:     **else**
18:        $\mathcal{L}_{tot} \leftarrow \mathcal{L}_{xe}(S_t; f_{\theta_t})$
19:        $\tau \leftarrow \text{Median}(E_t)$
20:     **end if**
21:     Compute the gradient $g_t$ of $\mathcal{L}_{tot}$
22:     $\theta_{t+1} \leftarrow \theta_t + \eta \cdot g_t$
23: **end for**
24: **return** a trained model $f_\theta$

---

The algorithm is an adaptation of the popular training method, mini-batch stochastic gradient descent (SGD), to our Hessian-aware training method. Any gradient-based training methods can be adapted to our Hessian-aware training. The changes we made are highlighted in blue.

In each mini-batch (line 3–22):

**(line 3–5, 20–21)** We compute the loss $\mathcal{L}$ of a model $f_{\theta_t}$ and update the model parameters $\theta_t$ with its gradient $g_t$. This step is the same as the original mini-batch SGD.

**(line 6–13)** This step computes the Hessian trace and eigenvalues with respect to the model parameters $\theta_t$. Computing the full Hessian matrix and its trace is computationally expensive than standard training (see Appendix B.7); we thus approximate the trace and eigenvalues using single step of the Hutchinson's method (Hutchinson, 1989), following the technique employed in prior work (Yao et al., 2020c; 2021a).

Suppose the Hessian $H \in \mathbb{R}^{d \times d}$ and random vector $v \in \mathbb{R}^d$ satisfying $\mathbb{E}[vv^T] = \boldsymbol{I}$. $v$ is drawn from Rademacher distribution which ensures half of the discrete probabilities are positive and the other half is negative ($P(v = \pm 1) = 1/2$). $d$ denotes the total number of parameters. In Hutchinson's method, the Hessian trace is calculated over a set of random vectors:

$$Tr(H) = \mathbb{E}[v^T H v] = \frac{1}{N_v} \sum_{i=1}^{N_v} v_i^T H v_i$$

where $N_v$ is the number of random vectors used to approximate. We can obtain $v^T H v$ by computing the gradient of the loss function $\mathcal{L}$ twice as follows:

$$v^T H v = v^T \cdot \frac{\partial}{\partial \theta}\left(\frac{\partial \mathcal{L}}{\partial \theta}\right) \cdot v$$

We follow the prior work (Yao et al., 2020c) to compute set of Top-$p$% eigenvalues $\lambda_p$ as follows:

$$\lambda_p = \frac{v_k^T H v_k}{\|v_k^T\|} \quad \text{for } k = 1, 2, \cdots, p$$

**(line 14–19)** In our experiments, we find that minimizing the Hessian trace computed on all eigenvalues can make the optimization process unstable. Instead, we take the $p$-largest eigenvalues to compute the trace. There will be negligible impact since the eigenvalues consist a few large values (representing the sharpest directions in the loss surface) and many smaller ones.

To identify an optimal $p$ value, we compare the effectiveness of computing only top-$p$ eigenvalues in minimizing a model's sensitivity to bitwise errors in parameters. Table 1 summarizes our findings. We train MNIST models and measure the sensitivity by computing the Hessian trace on a trained model. We observe that, when we use top-50% of the eigenvalues, this results in the higest average accuracy of 98.92% and the lowest sensitivity (86.94%). We thus use the top 50% of the eigenvalues for the rest of our paper. In addition to using the top 50% of

Table 1: **Comparing our method using the Hessian trace from Top-$p$ eigenvalues.** Each row reports the average mean and standard deviations of the traces we compute over 1000 random samples, repeated five times across five different models.

| Training Methods | Acc. | Sensitivity |
|---|---|---|
| **Baseline** | $98.55 \pm 0.53$ | $126.15 \pm 63.59$ |
| **Top-10% Eigenvalues** | $98.16 \pm 0.21$ | $128.58 \pm 61.85$ |
| **Top-25% Eigenvalues** | $97.96 \pm 0.22$ | $116.10 \pm 53.77$ |
| **Top-50% Eigenvalues** | $98.92 \pm 0.20$ | $86.94 \pm 38.93$ |

eigenvalues, we track the trace values over the course of training and only regularize the model when the trace computed for a mini-batch is greater than the average trace values observed previously. These two strategies we employ help stabilize our training and allowing us to achieve reasonable performance.

### 4.3 COMPARING WITH EXISTING APPROACHES TO MINIMIZING SHARPNESS

We next empirically evaluate and compare the effectiveness of our method with existing approaches to reducing the sharpness of a model during training. We compare our approach to three representative methods: (1) $\ell_2$-regularization, which has been shown empirically to reduce the sharpness of a model in literature (Foret et al., 2021); (2) AdaHessian (Yao et al., 2021a), a second-order optimizer demonstrated to be effective in reducing the sharpness; and (3) Sharpness-aware minimization (SAM) (Foret et al., 2021), a training method specifically designed to reduce the sharpness.

**Methodology.** We train MNIST and CIFAR10 model and measure the accuracy and sensitivity. For each model, we compute the Hessian trace five times on 1000 randomly chosen training samples. For each method, we run training five times and report the average. Because we empirically find that SGD struggles with optimizing our second-order objective across the hyperparameters we use, we train our models with the RMSProp optimizer unless otherwise specified. We choose the

Table 2: **Comparison to existing training methods.** We compare the accuracy and the sensitivity from existing approaches to our method. The metrics are computed across five different models, and the sensitivity are computed over 1000 samples randomly chosen from the training data.

| Training Method | MNIST | | CIFAR10 | |
|---|---|---|---|---|
| | Acc. | Sensitivity | Acc. | Sensitivity |
| Baseline | 98.90 | $123.68 \pm 63.79$ | 92.43 | $3808.91 \pm 803.19$ |
| L2-Regularization | 97.30 | $128.23 \pm 52.42$ | 91.72 | $4117.33 \pm 1032.42$ |
| AdaHessian | 98.88 | $126.67 \pm 70.82$ | 92.68 | $3717.55 \pm 931.80$ |
| SAM | 97.15 | $134.08 \pm 75.04$ | 92.15 | $3676.89 \pm 899.82$ |
| Hessian Trace (Min-max) | 98.65 | $128.72 \pm 68.50$ | 92.34 | $3571.88 \pm 924.67$ |
| Hessian Trace (Top-50, $\lambda$ to $10^{-4}$) | 98.78 | $126.67 \pm 70.82$ | 92.58 | $3543.33 \pm 952.44$ |
| Hessian Trace (Top-50, $\lambda$ to 1) | 98.92 | $86.94 \pm 38.93$ | — | — |
| Hessian Trace (Top-50, $\lambda$ to $10^{-2}$) | — | — | 92.71 | $2729.53 \pm 762.94$ |

learning rate and regularization coefficient $\lambda$ from {1, 0.1, 0.01, 0.001, 0.0001}, batch size from {32, 64}, and the number of Hutchinson's steps for trace approximation from {1, 50, 100, 1000}. Through extensive hyper-parameter search, we find that using only a single step to compute the Hessian trace is the most effective.

**Results.** Table 2 summarizes our results. We show that compared to existing approaches, our hessian-aware training is more effective in reducing a model's sensitivity. We also test two techniques to smooth out the Hessian regularization loss $Tr_t$ that is fluctuating over training epochs: (1) Min-max optimization: we normalize the loss based on the min and max values of the eigenvalues $E_t$ defined by this formula: $Tr_{t_{norm}} = Tr_t - min(E_t)/max(E_t) - min(E_t)$, where $Tr$ denotes the hessian

trace and $t$ denotes the current step; and (2) the technique that only considers the loss when its value is greater than the one $\tau$ observed in the previous epoch (see line 14–19 in Algorithm 1). We additionally use this computationally inexpensive approach to determine and compare the impact of regularization coefficient $\lambda$, and we show that setting $\lambda$ to one for MNIST and to $\lambda$ to $10^{-2}$ for Cifar10 achieves the lowest sensitivity. For the rest of our experiments, we use this training configurations.

## 5 EMPIRICAL EVALUATION

Our evaluation focuses on answering the following research questions: **(RQ 1)** How resilient are models trained with our algorithm to bitwise errors in parameters? **(RQ 2)** What characteristics of these models make them resilient to parameter variations? **(RQ 3)** Beyond this parameter resilience to bitwise errors, what benefits does our training approach offer? We evaluate comprehensively across three datasets and five different network architectures (see Appendix A for more details).

### 5.1 ENHANCED MODEL RESILIENCE TO BITWISE ERRORS IN PARAMETERS

**Methodology.** We first compare our approach with prior works and determined that ours is most effective in improving resilience against bitwise errors in parameters (see Appendix B.1). Then we perform a quantitative analysis of the enhanced resilience of a model to bitwise errors in parameters. We extensively examine a model's resilience under a single bitwise corruption in its parameter space because: (1) this approach allows us to test the most sensitive cases under atomic perturbations that can occur in models deployed to real-world devices, and (2) it also enables us to simulate the numerical value changes in all parameters under the smallest perturbation. We report the accuracy of a model to demonstrate that our training algorithm does not harm its generalization ability. We report the resilience of a DNN model by the number of erratic parameters that contain at least one bit whose error can result in an accuracy drop of over 10% as defined by (Hong et al., 2019). We also report their ratio to the total number of parameters. As the definition suggests, lesser erratic parameters in a DNN constitutes greater resilience against bitwise errors in parameters. For MNIST (Base and LeNet architecture), we test all bitwise errors in the model parameters and found that erratic bits are only present in the exponents (Appendix B.8). Complete analysis of the model requires infeasible computation time, for CIFAR model (ResNet18) ≈503 days and for ImageNet model (ResNet50) ≈1172 days. Based on our initial findings and the prohibitive computation time, we implemented techniques to accelerate the process. In CIFAR-10, we examine the exponent bits and found only few bits other than the MSB are responsible for RAD over 10%. For our ImageNet model, we test only the most significant bit of the exponent in a randomly chosen 50% of parameters in all the convolutions layers and all the parameters in the fully-connected layers.

Table 3: **Effectiveness of our Hessian-aware training.** We quantitatively compare the resilience of models trained with and without our approach to a single-bit error in their parameter space. *Ours* refers to the models trained with our approach, while *Baseline* is to the models trained without. Acc for both these models refer to the validation accuracy.

| Dataset | Model | Params | Bits | Baseline | | | Ours | | |
|---------|-------|--------|------|------|------------|-----------------|------|------------|-----------------|
| | | | | Acc. | Err. Params | Err. Param Ratio | Acc. | Err. Params | Err. Param Ratio |
| MNIST | **BaseNet** | 21,840 | 698,880 | 98.73 | 10,544 | 48.27% | 98.66 | 8,482 | 38.83% |
| | **LeNet** | 44,470 | 1.4M | 99.61 | 20,712 | 46.57% | 98.91 | 15,383 | 34.59% |
| **CIFAR-10** | **ResNet18** | 11M | 352M | 92.43 | 4.4M | 40.12% | 93.68 | 3.7M | 33.6% |
| **ImageNet** | **ResNet50** | 25.6M | 819.2M | 76.13 | 5,283,102 | 43.35% | 75.09 | 4,459,141 | 36.59% |

**Results.** Table 3 summarizes our results. We first find that our Hessian-aware training preserves the generalization ability. In all cases, the acc columns show that, there are negligible differences in Top-1 classification accuracy between the baseline models and the models trained with our method. More importantly, our approach reduces the number of erratic parameters by 5.2–11%: In MNIST and CIFAR-10 models, we observe a 10% reduction, while the reduction is 6.76% in the ImageNet model. We attribute this difference in reductions to the training strategy we employ for ImageNet-scale models. Instead of fine-tuning all layers of an ImageNet model, we focus only on the last fully-connected layer, which our analysis identifies as the most sensitive layer, to minimize the Hessian trace.

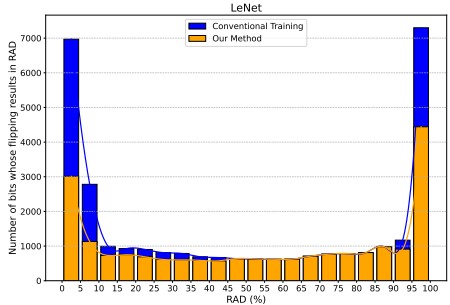 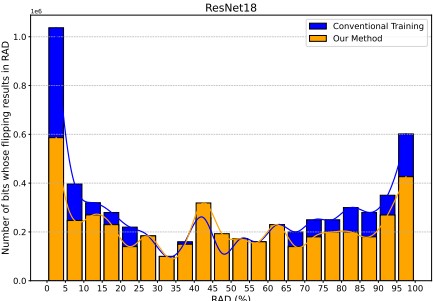

Figure 2: **The distribution plots illustrating the enhanced resilience of a model under parameter corruptions.** We compare LeNet trained on MNIST (left) and ResNet18 trained on CIFAR-10 (right).

To gain a deeper understanding of the enhance resilience by our approach, we compare the distribution plots between two models: one trained with our algorithm or the other without. Figure 2 illustrates the comparison. The figure shows the plots from the LeNet and ResNet18 models; the plots from the ImageNet models are provided in Appendix B.2. We analyze the distribution of accuracy drop from 0–100% in 5% increments. Across the board, our approach reduces the accuracy drop in two regions: (1) bits whose flipping leads to significant performance loss (90–100%) and (2) bits whose corruptions result in a small accuracy drop (0–10%). This implies that we reduce the chances of a model's performance becoming random due to bitwise errors in parameters by almost half.

## 5.2 CHARACTERIZATION OF THE ENHANCED MODEL RESILIENCE

We delve deeper into how various properties of a model affect its resilience to bitwise errors.

**Visualizing the loss landscape.** We first analyze whether the models trained with our method have a *flatter* loss surface than the baselines. We adopt the visualization technique proposed by Li et al. (2018): We choose two random vectors with the same dimension as that of a model's parameters and incrementally increase the perturbations to each direction to the parameters while measuring the loss value of the perturbed model. Figure 3 visualizes the loss landscape computed for each layer of the LeNet models trained on MNIST. From left to right, we visualize the five layers from the input.

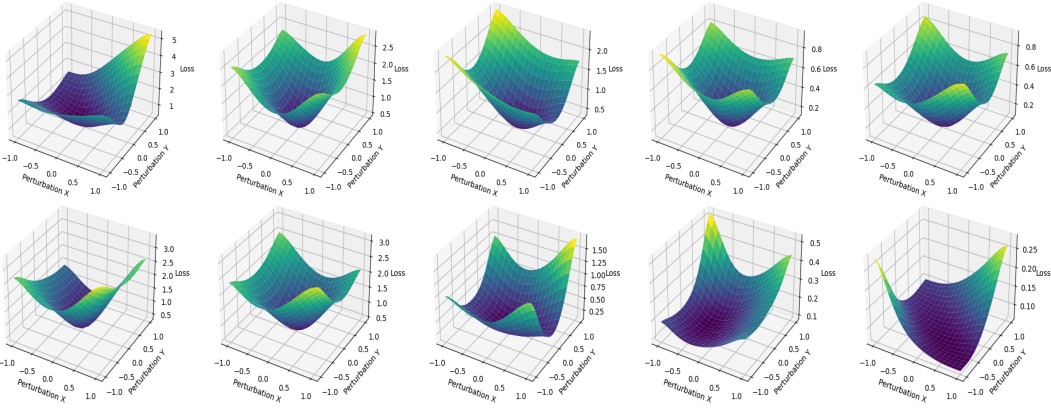

Figure 3: **Visualizing the loss landscapes of LeNet.** The upper row displays the loss landscapes of the baseline LeNet, while the lower row shows those of the LeNet trained with our method. From left to right, we visualize the first two convolutional layers followed by the three fully-connected layers.

We demonstrate that our training method effectively reduces the sharpness (i.e., the sensitivity of a model to bitwise errors) across all layers. Our approach is particularly effective in reducing the sharpness of the layers close to the output. In the last three columns of the figure, we observe that the loss curvatures become flatter compared to other layers. These three columns correspond to the fully-connected layers; thus, we further investigate whether fully-connected layers are particularly

less resilient without our training method (see the next paragraph). However, we also find that our approach is less effective at reducing the sharpness of the convolutional layers within the residual blocks (see Appendix B.3). This corroborates the observations made by Li et al. (2018) that residual connections offer a flat loss landscape. Our method may not offer significant resilience for those already-flat layers as the inherent flatness in their loss landscape is less sensitive to parameter perturbations. Our method is most impactful in sensitive layers (Conv and Fully Connected) of a DNN model, where the loss surface tends to be sharper.

Table 4: **Comparing the effectiveness of our approach in convolutional (Conv.) and fully-connected (FC) layers.** Ours refers to the models trained with our approach, while `Baseline` is to the models trained without. In Column 4, we show the # of parameters in Conv or FC layers, with the parenthesis indicating their ratio in each model. All other numbers show erratic parameters and their ratios as defined by Hong et al. (2019). The last two columns are the reduction in the two metrics.

| Dataset | Model | Layers | # Params | Baseline | | Ours | | Reduction | |
|---|---|---|---|---|---|---|---|---|---|
| | | | | #Err. Param | Ratio | #Err. Param | Ratio | #Err. Param | Ratio |
| MNIST | BaseNet | Conv. | 5,280 (24.2%) | 3,003 | 56.87% | 2,695 | 51.04% | 308 | 5.83% |
| | | FC | 16,560 (75.8%) | 7,544 | 45.55% | 5,811 | 35.09% | 1,733 | 10.46% |
| | LeNet | Conv. | 2,616 (5.9%) | 1,719 | 65.71% | 1,475 | 56.38% | 244 | 9.33% |
| | | FC | 41,854 (94.1%) | 20,013 | 47.81% | 14,903 | 35.61% | 5,110 | 12.20% |
| CIFAR-10 | ResNet18 | Conv. | 11.2M (99.7%) | 4.4M | 40.07% | 3.7M | 33.57% | 0.7M | 6.50% |
| | | FC | 5,120 (0.03%) | 2,297 | 44.86% | 1,321 | 25.80% | 976 | 19.06% |
| ImageNet | ResNet50 | Conv. | †23.5M (53.5%) | 4,516,162 | 38.23% | 3,802,648 | 32.19% | 713,514 | 6.04% |
| | | FC | 2.04M (46.5%) | 766,940 | 37.59% | 656,493 | 32.18% | 110447 | 5.40% |

**Resilience of convolutional layers vs. fully-connected layers.** Our previous analysis of the loss surfaces shows that our approach tends to reduce the sensitivity (i.e., sharpness) of the later layers. Since most feed-forward neural networks have convolutional layers followed by fully connected layers for classification, we analyze whether the resilience has indeed increased in the fully connected layers. Table 4 summarizes our findings. Across all models, we observe that the reduction in the ratio of erratic parameters in fully connected layers is 2.4–13.4% greater than that in convolutional layers. Particularly, for the ResNet18 trained on CIFAR-10, our Hessian-aware training reduces the erratic parameter ratio by 19.1%. This result implies that network architectures with many fully connected layers, such as BaseNet or LeNet, can benefit more from our method. But architectures like ResNets, composed of 99% of convolutional layers (that is a feature extractor) followed by one or two fully connected layers, may experience a reduced benefit. Given the recent paradigm shift from feed-forward convolutional networks to Transformers composed of many fully-connected layers like ViTs (Dosovitskiy et al., 2021), we believe that it is important to evaluate whether our training method will be effective for these models. We apply our training method to fine-tune Deit-Tiny (Touvron et al., 2021), a Transformer-based model, pre-trained on ImageNet (available at HuggingFace[1]) and measure the improvement in resilience. Due to the page limit, we present our results on DeiT in Appendix B.4. Our method reduces the number of erratic parameter ratio from 43.7% to 36.8%.

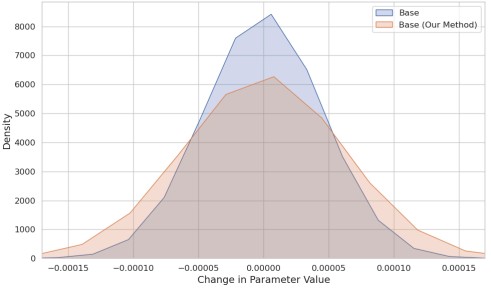

Figure 4: **Comparison of numerical variations required to cause a RAD drop over 10%.** We compare the distributions of numerical perturbations to parameters that are needed to cause significant performance loss (RAD > 10%).

**Resilience to parameter value changes.** We lastly analyze how resilient a model becomes to actual parameter value changes caused by single bitwise errors. Using the parameter values before any corruption and after causing a single-bit error, we compute the changes in the numerical values. Figure 4 shows our analysis results on LeNet in MNIST. Due to the space limit, we include the rest plots in Appendix B.5. We demonstrate that DNN models trained with our method requires a greater numerical variations to cause a RAD drop over 10% than those trained using regular training methods. Based on our observation (see Appendix B.5) that most single-bit errors cause a bit-flip in the most significant bit of the exponent (i.e., the 31st-bit), the numerical

---

[1]DeiT-tiny: https://huggingface.co/facebook/deit-tiny-patch16-224

variations required to cause a large performance loss go beyond the range that floating-point representation in modern systems can hold.

### 5.3 ENHANCED MODEL RESILIENCE TO COMPRESSION

We now examine the additional benefits of our approach beyond parameter resilience to bitwise errors. We are particularly interested in testing whether models trained with our method can achieve improved performance under pruning (Han et al., 2015) or quantization (Fiesler et al., 1990). These techniques reduce the size of neural networks through parameter reduction or compression, introducing optimal parameter perturbations (LeCun et al., 1989). It is important to study the effectiveness of our method in increasing the resilience of DNN models against these perturbations.

**Pruning.** In our evaluation, we employ global unstructured pruning (Liu et al., 2017), which operates at the individual weight level. This technique first computes an importance score for each weight and removes those with the lowest scores. We apply this pruning technique with different sparsity levels ranging from 0–100%. Figure 5 shows our pruning results on the ResNet18 models trained on CIFAR-10. The rest of our results are in Appendix B.6. We demonstrate that DNN models trained with our method retain accuracy better than those trained using regular training methods. Both models retain their original accuracy up to the point where 50% of the parameters are pruned. Our approach surprisingly maintains accuracy further, up to 70% pruning, while at the same sparsity level, the model trained with the conventional approach completely loses accuracy (i.e., the accuracy dropping to ∼0%).

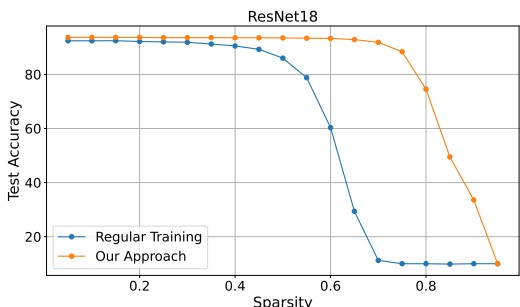

Figure 5: **Comparison of model performance under various pruning ratios.** We compare the test accuracy of the ResNet18 models trained on CIFAR-10. The magnitude-based iterative pruning is used to achieve sparsity levels from 0–100%.

**Quantization.** Table 5 summarizes our quantization results for 8-, 4-, and 2-bit quantization of the regularly-trained models and Hessian-aware trained models. We employ layer-wise, symmetric quantization, which is the default in most deep learning frameworks. Overall, the models trained with our approach achieve better test accuracy than the regularly trained models, an additional benefit that hessian-aware training offers. Up to 4-bit quantization, both models retain the performance of their floating-point counterparts. However, when we use 2-bit precision, the accuracy of all models decreases significantly. Our models under 2-bit precision consistently achieve 1.5–14% better accuracy, indicating that these models have increased resilience to parameter value variations. Based on our observation that fully-connected layers are less sensitive than convolutional layers (see the above analysis), we employ mixed-precision quantization with 2-bit precision in fully-connected layers and 4-bit precision in convolutional layers. We demonstrate that our models achieve an accuracy of 68.8–78.7%, while the regularly-trained models achieve 48.9–68.2% model accuracy.

Table 5: **Comparison of model performance under various quantization ratios.** We compare the test accuracy of models after quantizing them with different bit-widths.

| Dataset | Model | Acc. | | | |
|---|---|---|---|---|---|
| | | **8-bit** | **4-bit** | **2-bit** | **Mixed** |
| MNIST | **Base** | 98.57 | 98.38 | 24.49 | 48.90 |
| | **Base (Ours)** | 98.73 | 98.70 | 38.72 | 68.84 |
| | **LeNet** | 99.10 | 98.70 | 11.85 | 57.03 |
| | **LeNet (Ours)** | 98.90 | 97.37 | 24.78 | 73.90 |
| CIFAR-10 | **ResNet18** | 92.53 | 88.01 | 9.96 | 68.19 |
| | **ResNet18 (Ours)** | 92.36 | 90.26 | 10.28 | 78.69 |

## 6 DISCUSSION

**Increase in computational demands.** We evaluate the overhead of Hessian-aware training in terms of actual training wall-time measured in PyTorch on a NVIDIA Tesla V100 GPU. In Appendix B.7

we present our results. Hessian-aware training incurs overhead that scales with the size of the model; a 4–6$\times$ times increase in computations for MNIST models, and a 10$\times$ times increase in overhead for CIFAR-10 models. Existing works utilizing second-order properties during training take a completely different approach compute the Hessian and its eigenvalues: they employ weight perturbations (Foret et al., 2021) or only the trace approximation (Yao et al., 2021a) to minimize sharpness of the loss landscape. The increase in computation in our approach is primarily attributed to the large Hessian and eigenvalues we need to compute with respect to model parameters, which is not optimized for popular deep-learning frameworks. To reduce the computational overhead during training, we employ a layer sampling technique. As prior work identifies the last layers to be most susceptible to bitwise errors (Hong et al., 2019), we believe only computing Hessian trace on last few layers can aid resilient model training. Our results for large-scale models, such as ResNet50 in ImageNet, show that this technique significantly reduces the computational overhead from 10$\times$ times to 1.18$\times$ times, being equally effective in enhancing model resilience. We leave further optimization as future work.

Now we discuss defensive mechanisms proposed by the community that can be integrated with our approach to further enhance the resilience of models against bitwise errors in parameters. Our discussion particularly focuses on hardware-level and system-level approaches, which can complement the models trained with our Hessian-aware training by providing an additional layer of resilience.

**Integration with system-level defenses.** Liu et al. (2023) have proposed NeuroPot, which lures an adversary into manipulating parameters whose perturbation does not lead to a significant accuracy drop. Since our approach increases the number of unimportant parameters, the models we train are well-suited for NeuroPot. Li et al. (2021) have proposed a checksum-based defense, which stores *golden signature* for a group of weights and compares this signature at runtime with the current model signature. Our approach can leverage this scheme by storing the golden signature of erratic weights, thus enhancing resilience further during runtime. Konoth et al. (2018) have also focused on utilizing data row isolation to protect a model against bitwise corruption to its parameters. Because we reduce the number of erratic parameters, the number of data rows that need to be isolated in their approach can also be greatly reduced when combined with our training method.

**Integration with hardware-level defenses.** Many hardware-level defenses are designed to mitigate RowHammer (Kim et al., 2014), a software-induced attack that causes a targeted DRAM row to leak capacitance by repeatedly accessing its neighboring rows. Kim et al. (2014) have proposed a defense that proactively refreshes rows that are frequently accessed, as they are at higher risk of being targeted by the attack. Panopticon (Bennett et al., 2021) leverages a similar idea: it employs hardware counters for each data row in DRAM and refreshes the rows when the counter reaches a predefined threshold. Instead of refreshing the rows at high risks, Saileshwar et al. (2022) propose swapping them with safe memory regions. Di Dio et al. (2023) use the error correction codes as a mechanism for triggering such swapping. DRAM-Locker (Zhou et al., 2023b) leverages a lock-table in SRAM. If the addresses of the high-risk rows are stored in the lock-table, any access this addresses without the unlock command will be denied. As we can see, these mechanisms protect data rows at high risk of being targeted. Our work reduces the number of data rows in a model whose perturbations lead to significant accuracy loss, and therefore, potentially decreasing their performance overheads.

## 7 CONCLUSION

Our work proposes a novel training algorithm that reduces a model's sensitivity to parameter variations, thereby enhancing its resilience when deployed in error-prone computing environments. We focus on the model's second-order property, the Hessian trace, and design an objective function to directly minimize it during training. We extensively compare our approach with existing methods for reducing model sensitivity and demonstrate our effectiveness. We evaluate our approach by testing a model's performance under single-bit errors to its parameter representation in memory. Our method reduces the number of erratic parameters by 10%, decreasing those whose corruption causes a 90–100% RAD drop by almost half. Our method is particularly effective on fully connected layers, and results in flatter loss surfaces. We also show improved performance at an extreme pruning and quantization ratios. Our method is complementary to existing hardware-level or system-level approaches to protecting models against parameter-corruption attacks. We finally discuss the potential synergy of combining these mechanisms with our method. We hope our work will inspire future work on the safe deployment of deep neural networks in emerging computing platforms.

**Reproducability Statement.** To make our work reproducible, we provide description of the dataset, models, hyper-parameters and our Hessian-aware training method both in the main text and in Appendix. Specifically, Sec 3 and Appendix A offer detailed discussion on our models, datasets and training hyper-parameter settings. We discuss our proposed hessian-aware training algorithm in Sec 4.2. We believe these detailed implementation descriptions will facilitate the successful replication of our work. We will also release the source code to further ensure the reproducibility.

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

## A    DETAILED EXPERIMENTAL SETUP

Here we describe our experimental setup in detail. All experiments use Python v3.11.4[2] with Pytorch v2.1.0[3] and CUDA v12.1[4] for GPU acceleration. We run our experiments on two systems: (1) a node with a 48-core Intel Xeon Processor, 768GB of memory, and 8 NVIDIA A40 GPUs. (2) a node with a 56-core Intel Xeon Processor, and 8 Nvidia Tesla H100 GPUs. We achieve a substantial speed-up in running our evaluation script by utilizing the parameter-level parallelism on the two systems.

We use the following hyper-parameters to train/fine-tune our models.

**MNIST.** We use a network architecture (Base) and LeNet in prior work (Hong et al., 2019). For regular training, we used an SGD optimizer with a learning rate of 0.1 (adjusting by 0.25 every 10 epochs), batch size of 64, and 0.8 momentum. We train our models for 40 epochs. To train the same network using our Hessian-aware training, we used $\lambda$ (line 16 of algorithm 1) value of 1 as per our findings in table 2. We use the RMSProp optimizer, keeping all the other hyper-parameters the same as the regular training.

**CIFAR-10.** We use ResNet18. For the regular training of this model, we use SGD, 0.02 learning rate, 32 batch-size, 0.9 momentum. We train our models for 90 epochs. We adjust the learning rate by 0.5 every 15 epochs. We use the RMSProp optimizer and $\lambda$ value of $10^{-2}$ to train the same model with our training method.

**ImageNet.** We take the ResNet50 architecture pretrained on ImageNet (available at Torchvision library[5]). Instead of retraining the ResNet50 from scratch, we fine-tune the model on the same ImageNet dataset. In fine-tuning, computing the Hessian matrix has a high computational demand. We thus leverage our previous observation and focus on the layers closer to the model output. We only compute Hessian eigenvalues and trace on the last layer and fine-tune the entire model using our training method. The hyper-parameters have been kept as Torchvision's original training hyper-parameters [6]), but using the RMSProp optimizer. For fine-tuning the Diet-tiny ViT model on ImageNet, we use similar technique for hessian and eigenvalue computation. We take the pre-trained model from HuggingFace (available at [7]) and fine-tune it using our approach. We adopt the original training setup from (Touvron et al., 2021), that uses batch size of 32, learning rate 0.1 and reducing by 0.1 every 30 epoch, momentum of 0.9, weight decay $10^{-4}$ and 90 epochs training cycle except we use the RMSProp optimizer. We experimentally found $\lambda$ value of $10^{-3}$ to achieve better generalization for our ImageNet model.

## B    ADDITIONAL EVALUATION RESULTS

### B.1    COMPARING THE RESILIENCE IN OUR APPROACH AND PRIOR WORKS

Table 6: **Comparing resilience of our approach with prior works on second-order methods.** The Base model is trained on MNIST using AdaHessian (Yao et al., 2021a), SAM (Foret et al., 2021), and our method. Column 5 reports the number of erratic parameter and column 6 their ratio to the total model parameters.

| Training Method | Model | # Total Params | Acc. | Erratic Params | Erratic Ratio |
|---|---|---|---|---|---|
| **AdaHessian** | Base | 21,840 | 98.88% | 10,473 | 47.72% |
| **SAM** | | | 97.15% | 10,621 | 48.63% |
| **Our** | | | 98.66% | 8,482 | 38.83% |

In Sec 5.1 we discuss our results on enhanced model resilience to bitwise errors in parameters. Here we compare the resilience of our approach to prior works on sharpness minimization using second

---

[2]Python: https://www.python.org

[3]PyTorch: https://pytorch.org/

[4]CUDA: https://developer.nvidia.com/cuda-downloads

[5]Pre-trained PyTorch models: https://pytorch.org/vision/stable/models.html

[6]https://github.com/pytorch/vision/tree/main/references/classification

[7]DeiT-tiny: https://huggingface.co/facebook/deit-tiny-patch16-224

order method, SAM (Foret et al., 2021) and AdaHessian (Yao et al., 2021a). We measure resilience in terms of the the number of erratic parameters and their ratios as defined in section 5.2. Table 6 shows our results. We find that our hessian aware training is more effective in enhancing resilience in a DNN model. This finding is consistent with the results in table 2, where we show our approach being the most effective in reducing sensitivity.

## B.2    Distribution Plot Computed on ImageNet Models

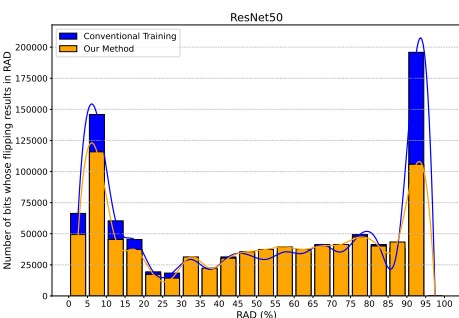

Figure 6: **The distribution plot computed on ResNet50 in ImageNet.** Note that our fine-tuning only computes the Hessian trace from the last layer.

We show the distribution plot computed on the ImageNet models in figure 6. We observe that fine-tuning the pre-trained ResNet50 achieves an enhanced resilience to bitwise errors in parameters. It reduces the number of corruptions leading to an accuracy drop in the range between 0-30%. We also reduce the number of parameters whose bitwise error leads to an accuracy drop of over 90% by half. Our result on ImageNet is particularly interesting because, even if we do not train our model with the Hessian trace computed on the entire layers, we can offer enhanced resilience to a model. While in MNIST and CIFAR-10 models, we see the number of parameters causing accuracy loss of 0–5%, in our fine-tuned ImageNet model, we find a greater number of parameters causing accuracy drops at 5–10% bin.

## B.3    Visualizing Loss Landscapes of Layers with Residual Connections

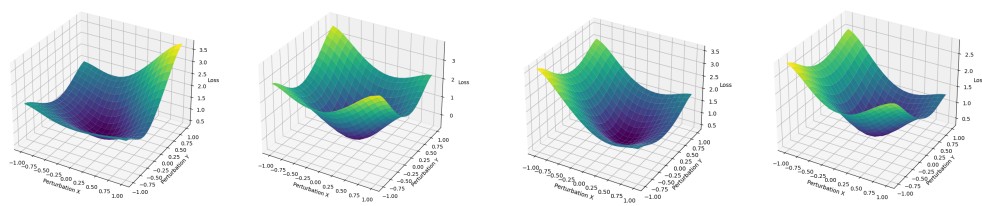

Figure 7: **Comparing loss landscapes of the convolutional layers within a residual block.** The left two are from the regularly-trained models, and the right ones are from those trained with our method.

Prior work (Li et al., 2018) has visually shown that convolutional layers with residual connections tend to have flatter loss surfaces. In such layers, we hypothesize that our approach is less effective in reducing the sensitivity. Figure 7 shows the loss landscapes from two convolutional layers in ResNet18 models trained on CIFAR-10. We observe that the loss landscapes visually look similar to each other, implying that our approach was less effective in reducing the Hessian trace of these layers. This does not mean that these layers are particularly susceptible to bitwise errors in parameters. On the other hands, these convolutional layers already have some resilience to bitwise errors in parameters.

## B.4    Effectiveness of our approach in Visual Transformer model

Evaluating our approach on Transformer-based models presents an interesting extension of this work. Our initial hypothesis is that, in the context of computer vision, Transformer layers function as an extension of fully-connected layers. Images are split into patches, flattened, and linearly projected (via a fully-connected layer) to create patch embeddings, which are processed by a transformer encoder using multi-head self-attention and feedforward networks (composed of fully-connected layers), and the [CLS] token is passed through a final fully-connected layer for classification tasks (Dosovitskiy et al., 2021). This hypothesis leads us to anticipate that the outcomes will align with our current findings. Our results are shown in table  7.

Table 7: **Comparing the effectiveness of our approach in enhancing resilience of visual transformer (ViT) model.** We take the `Baseline` model pretrained on ImageNet. `Ours` refers to the model fine-tuned with our approach. We report the resillience in terms of erratic parameter in both these models.

| Model | # Total Params | # Sampled Params | Err. Params | Err. Param Ratio |
|---|---|---|---|---|
| **DeiT-tiny (Baseline)** | 5M | 457,000 | 199,571 | 43.67% |
| **DeiT-tiny (Ours)** | | | 168,358 | 36.84% |

We fine-tune the last layer of the Diet-tiny model following the original hyper-parameter setup (Touvron et al., 2021). We ran our resilience measurement analysis as defined in section 5.2. The results demonstrate the effectiveness of our training method: the number of erratic parameters was reduced by 6.83%, which is consistent with the results in our paper across different datasets and models.

### B.5 NUMERICAL PERTURBATIONS CAUSING ACCURACY DROP OVER 10%

We examine the change in parameter values after a single bit corruption on the two Base models (one regularly-trained, and the other trained with our approach). Figure 8 shows our results. We show our results from the LeNet and ResNet18 models. Similar to our previous finding in the Base model, larger models (LeNet and ResNet18) trained using our method also demonstrate parameter-level resilience. To cause an accuracy drop over 10%, model trained using our method requires greater numerical perturbations.

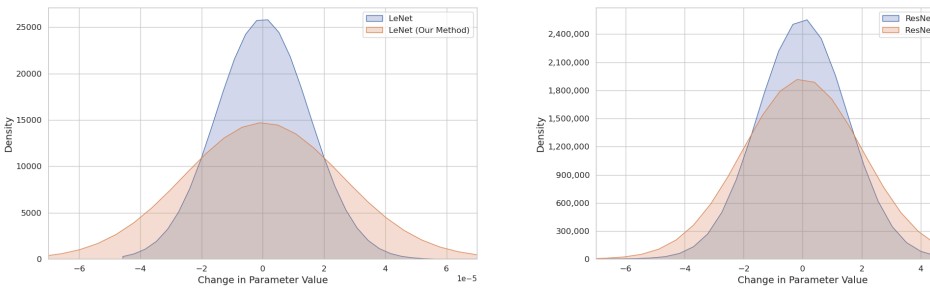

Figure 8: **Comparison of numerical perturbations required to cause an accuracy drop over 10%.** The left figure is computed on ResNet18, and the right one shows the result of LeNet.

### B.6 ADDITIONAL PRUNING RESULTS

In section 5.3, we discuss the effectiveness of our Hessian-aware training in achieving DNN models resilient to model compression in CIFAR-10. Here we show the results of pruning on our MNIST models, specifically Base and LeNet in figure 9. Both models retains their original accuracy up to 65% parameters pruned. Beyond this point, as sparsity increases, we observe a steep decrease in accuracy. The Base and LeNet models trained using our method shows better accuracy than the regularly-trained models, indicating enhanced parameter-level resilience to bitwise errors.

### B.7 OVERHEAD OF HESSIAN AWARE TRAINING

For all the models, we used the optimal hyper-parameter setup described in Appendix A. We run training for 5 times, and report the per epoch training time. The measured values are presented in Table 8. Result shows that hessian aware training has a 4x-6x overhead for MNIST model which have comparatively smaller number of parameters. For the ResNet18 model trained on CIFAR-10, the overhead increases further due to ResNet18's larger architecture and higher number of parameters compared to smaller models. We employ layer sampling technique to reduce this overhead. Prior research (Hong et al., 2019) suggests that these final layers are the most susceptible layers against parameter corruption, making this a viable strategy for applying our method to large-scale models,

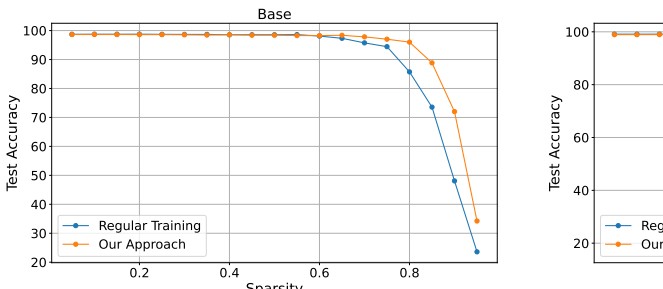

Figure 9: **Comparison of model performance under various pruning ratios.** The left figure is computed on the Base models, while the right ones are from the LeNet models.

Table 8: **Comparing the training time of our method to baseline training in terms of runtime in PyTorch.** We report the per-epoch runtime (in seconds) for all our models trained across 3 datasets.

| Model | Dataset | Training Time | |
|---|---|---|---|
| | | **Baseline** | **Our Method** |
| **Base** | MNIST | $0.335 \pm 0.002$ | $1.362 \pm 0.0085$ |
| **LeNet** | | $0.432 \pm 0.003$ | $2.857 \pm 0.0073$ |
| **ResNet18** | CIFAR10 | $36.244 \pm 0.607$ | $341.58 \pm 9.81$ |
| **ResNet50** | ImageNet | $7275.6 \pm 18.41$ | $8647.2 \pm 25.43$ |

such as those used for ImageNet. Our result shows that adopting this method has only 1.18x computational overhead. We conduct additional experiment on the layer-sampling technique for larger architecures like ResNet18 and ResNet50. Following the same overhaed measurement approach, We applied Hessian regularization incrementally, starting with only the last layer and extending it to the last 2, 3, and finally 4 layers of the model and compared the runtime with baseline training. Our results are presented in Table 9.

Table 9: **Comparing the training time of layer-sampling and baseline training in PyTorch.** We report the per-epoch runtime (in seconds).

| Model | Dataset | Training Time (in seconds) | | | | |
|---|---|---|---|---|---|---|
| | | **Baseline** | **L1** | **L2** | **L3** | **L4** |
| **ResNet18** | CIFAR10 | $36.244 \pm 0.607$ | $37.77 \pm 0.39$ | $43.24 \pm 0.28$ | $57.63 \pm 0.44$ | $78.24 \pm 1.13$ |
| **ResNet50** | ImageNet | $7275.6 \pm 18.41$ | $8647.2 \pm 25.43$ | $10134.7 \pm 30.21$ | $13289.5 \pm 35.76$ | $16547.8 \pm 42.15$ |

Results in Table 9 demonstrate that training overhead increases as we increase the "layers involved in hessian-regularization." However, using only the last 1 layer of the model, we can reduce the overhead to almost the same as baseline training, making our method efficient for very large models. We note that the increased computational time is not solely due to adopting our training method. The additional time is primarily attributed to the computation of the large hessian trace and eigenvalues, which is not fully optimized for use with popular deep learning frameworks such as PyTorch. Further optimization of our approach will be an interesting future work.

## B.8 ANALYSIS OF CORRUPTED BIT POSITION

The IEEE 754 standard defines the representation of floating-point numbers in modern computer systems. In this format, a 32-bit number is represented with three fields: the 1-bit sign, the 8-bit exponent, and the 23-bit mantissa. Similar to the prior work (Hong et al., 2019; Rakin et al., 2019; Yao et al., 2020b), we analyze the location of bitwise corporations that lead to an accuracy drop over 10%. Figure 10 shows our analysis results. We use a logarithmic scale in the y-axis.

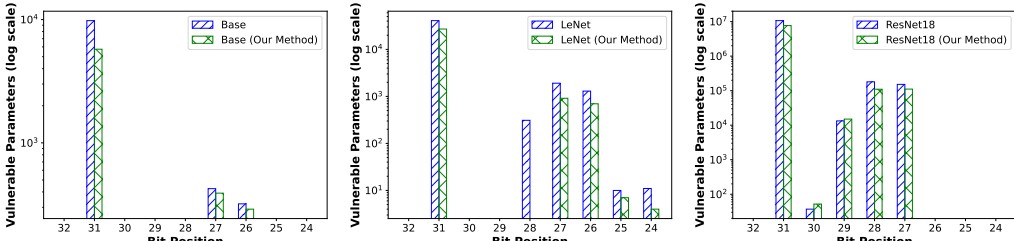

Figure 10: **Comparison of the corrupted bit positions.** From left to right, we show the analysis result from Base (MNIST), LeNet (MNIST), and ResNet18 (CIFAR-10). We only examine the sign bit and the exponent bits, as they change the numerical value of a parameter the most.

In all the models, corruption of the $31^{st}$ bit mostly leads to an accuracy drop over 10%. These corruptions account for $\sim$93% and $\sim$91.43% in the Base and LeNel models, respectively. We also observe a few bits in the $26^{th}$ and $27^{th}$ position for both Base and LeNet models and a small number of bits in the $28^{th}$ location for the LeNet model. A consistent trend is observed in the ResNet18 models in CIFAR-10, with the $31^{st}$ bit being identified as the most susceptible bit location. However, in ResNet18, we identify a few bits positioned at the $30^{th}$ and $29^{th}$ location in the exponent. In contrast to our observations from LeNet and ResNet18, there are no susceptible corruptions in the $30^{th}, 29^{th}, 28^{th}, 25^{th}$ and $24^{th}$ bit positions in the Base model.

