# OpenReview forum: "Hessian-Aware Training for Enhancing Model Resilience for In-Memory Computing"
_ICLR.cc/2025/Conference — Submitted to ICLR 2025_

### Official Review · Reviewer_2ZTo · 2024-10-30

**Soundness:** 2
**Presentation:** 3
**Contribution:** 2
**Rating:** 3
**Confidence:** 4

**Summary:**

This paper proposes a technique to improve resilience to bitwise errors. It consists of using the trace of the Hessian as a proxy for error sensitivity and, consequently, as an additional regularization term during model training. This methods results in models exhibiting flatter loss landscape as well as lower accuracy degradations when subject to adversarial bitwise flipping. Impact on pruning and quantization is also investigated. Results are reported on image classification tasks on a few models (BaseNet, LeNet, ResNet18, ResNet50) and datasets (MNIST, CIFAR-10, ImageNet).

**Strengths:**

- the paper is well structured and easy to follow

**Weaknesses:**

- the presentation of this paper is extremely misleading: it is introduced as a work on noise-resilience targeted at In-Memory Computing (IMC), but the methodology it employs for validation (adapted from [1]) applies specifically to adversarial attacks. The same adversarial attacks would have catastrophic impact on model performance irrespective of the hardware, IMC or not, when flipping a single exponent bit may change a value by several order of magnitude (depending on the precision being used). In fact, the original paper [1] deals with *non-IMC* models, on standard digital hardware. Noise characteristics typical of IMC hardware (e.g., Analog IMC) do not follow at all the patterns of adversarial attacks. While it is certainly the case that IMC may introduce additional sources of noise, the "frequent bitwise error" mentioned in the abstract is not uniformly distributed across bits. In this manuscript, the authors evaluate performance drop due to bitwise errors following 3 methodologies:
1. across all possible individual bits flipping for MNIST (giving equal weight to each configuration)
2. across only the exponent bits (thus the most impactful) for CIFAR-10
3. only changing the most significant exponent bit for ImageNet

None of these methods is a fair representation of noise in IMC hardware. They may be however suitable to analyze the impact of targeted adversarial attacks, as [1] does. My view is further supported by the fact that beyond title, abstract, and introduction, there is no mention of IMC-related experiments across the manuscript
- viewed as a manuscript about model safety / security instead of noise resilience for IMC, as hinted in two paragraphs in Section 6, the scope and potential impact become significantly more limited
- Fig. 1 and 2 show distributions plots of number of bits that cause a certain level of Relative Accuracy Drop (RAD). I would have expected to observe a shift in distribution but the integral below the updated curve (yellow) is significantly lower that the reference (blue). Does this mean fewer flip combinations were tested? Or does this mean instead that a significant percentage of bit flips caused negative RAD (i.e., improve performance) upon application of this method, and are therefore not included in this plot? This would be rather surprising
- the impact of training with Hessian-based regularization on pruning resilience is arguably the clearer demonstration of a potential advantage of this technique. However, the experimentation is limited to a single model/task combination, making it impossible to determine whether this outcome is generalizable
- quantization results are not impressive, with similar performance with vs. without Hessian regularization down to 4 bits. I would also argue that 4-bit weight-only quantization of such small models is a solved and currently irrelevant problem. Today, LLM with billions of parameters can quantized to 4 bits weights with minimal accuracy degradation. The 2-bit quantization results show some improvement, but the degradation observed is still massive, the models remain practically unusable


[1] Sanghyun Hong, Pietro Frigo, Yigitcan Kaya, Cristiano Giuffrida, and Tudor Dumitras. Terminal brain damage: Exposing the graceless degradation in deep neural networks under hardware fault attacks. In 28th USENIX Security Symposium (USENIX Security 19), pages 497–514, 2019.

**Questions:**

- I would recommend the authors to either re-frame the paper as a technique to mitigate adversarial attack, or present experiments that relate to IMC noise

---

> ### Author Response · Authors · 2024-11-25
>
> #### We thank reviewer 2ZTo for the time and effort in reading and evaluating our manuscript. Below, we answer the reviewer’s concerns and questions.
>
> ---
>
> **[Concern 1: Representation of noise]**
> #### We appreciate the reviewer’s concerns regarding the representation of noise specific to In-Memory Computing (IMC) hardware. We respectfully disagree with the reviewer and want to clarify that our study is more concerned about improving the parameter-level robustness of a DNN against bitwise error in model parameters. IMC noise models often apply random noise across parameters and report average performance losses, offering a general view of model resilience under typical noise conditions [A, B, C]. However, such average-case models may obscure cases where individual bitwise errors cause catastrophic accuracy drops, which is critical for robustness in fault-prone hardware environments like IMC. If we randomly apply noise to parameters, we barely find cases of accuracy drop. Our approach aims to identify and quantify worst-case scenarios, showing the impact of specific parameter manipulations on model accuracy. This allows us to understand and address parameters especially sensitive to bitwise faults, regardless of hardware type. On top of that, by systematically evaluating the sensitivity of each parameter to single-bit flips, we provide a more granular resilience analysis.
>
> ---
> **[Concern 2 : Clarification on figure 1 and 2]**
> #### A significant percentage of bit flips result in a negligible improvement in accuracy for Hessian-aware trained models. For example, a model with hessian-aware training with 92.67% baseline validation accuracy might exhibit a 92.68% validation accuracy following particular bit flips, resulting in a slight negative RAD score. Figures 1 and 2 do not include these instances, as the RAD metric only reflects more than 10% accuracy drops.
>
> ---
> **[Concern 3: Generalizability of our approach in pruning applications]**
> #### We appreciate the reviewer’s recognition of the advantages of Hessian-based regularization in enhancing pruning resilience. We would like to clarify that, contrary to the reviewer's impression, we conducted evaluations across multiple model and dataset combinations. These additional results, which support the generalizability of our approach in pruning applications, are included in the appendix section (B.6) of the manuscript. Our analysis demonstrates consistent pruning resilience benefits across different models and tasks, underscoring the broad applicability of our Hessian-aware training technique.

---

> > ### Author Response · Authors · 2024-11-25
> > **Rebuttal - cont'd**
> >
> > #### ...cont'd from the rebuttal
> > ---
> > **[Concern 4 : Concerns Regarding Quantization Results]**
> > #### We appreciate the reviewer’s feedback regarding our quantization results and would like to clarify the objective and relevance of our evaluation. We acknowledge that accuracy degradation remains a practical challenge, and our method offers incremental improvement rather than complete resolution. We used the most basic quantization technique to demonstrate our method's additional benefits in quantized models. Using the advanced quantization method, we expect to see better accuracies, however, it is outside the scope of the motivation and goal of our study. Achieving stability at extreme quantization levels demonstrates the robustness potential of our hessian-aware training, an area that remains largely unexplored. Our results lay the groundwork for future developments.
> >
> > ---
> > **[Summary]**
> >
> > #### We thank the reviewer for their insightful feedback and have addressed their concerns comprehensively. We clarified that our noise resilience analysis focuses on identifying worst-case bitwise errors rather than average-case IMC noise, providing a more granular perspective. Additionally, we explained the RAD metric used in Figures 1 and 2 and highlighted our method’s resilience benefits for pruning and extreme quantization. We kindly ask the reviewer to update his rating if his concerns have been addressed. We will be happy to address any remaining concern(s) during the discussion phase.
> >
> > ---
> > **[References]**
> > #### A. Shah, Vivswan, and Nathan Youngblood. "AnalogVNN: A fully modular framework for modeling and optimizing photonic neural networks." APL Machine Learning 1.2 (2023).
> >
> > #### B. Peng, Xiaochen, et al. "DNN+ NeuroSim V2. 0: An end-to-end benchmarking framework for compute-in-memory accelerators for on-chip training." IEEE Transactions on Computer-Aided Design of Integrated Circuits and Systems 40.11 (2020): 2306-2319.
> >
> > #### C. Rasch, Malte J., et al. "A flexible and fast PyTorch toolkit for simulating training and inference on analog crossbar arrays." 2021 IEEE 3rd international conference on artificial intelligence circuits and systems (AICAS). IEEE, 2021.

---

> > > ### Comment · Reviewer_2ZTo · 2024-11-25
> > > **response to rebuttal**
> > >
> > > I thank the authors for their response to my comments and those of the other reviewers.
> > >
> > > In reference to the main concern I expressed, I remain strongly convinced that presenting this investigation as a study on noise resilience in IMC would be extremely misleading.
> > >
> > > To reiterate, when hardware noise is discussed in IMC literature (including the 3 references the authors cited in their response), it is in reference to errors driven by hardware non-idealities, which result in read/write errors of model parameters and approximate computations. The distribution of such errors greatly differ from those investigated by the authors in this manuscript.
> > >
> > > I am puzzled by the authors' statement that "randomly apply noise to parameters, we barely find cases of accuracy drop". This statement may have limited validity in some specific experiments the authors may have carried out, but it is well known that, hardware noise (both simulated and measured) leads to dramatic degradation to model performance in a variety of scenarios, a degradation which calls for specific compensation strategies (see for example [1]).
> > >
> > > I challenge the notion that addressing the impact of individual bitwise errors is "critical for robustness in fault-prone hardware environments like IMC". I am willing to change my mind would the authors bring forward evidence that individual bitwise flipping, not only the hardware noise that is typically modeled and compensated, is an actual challenge that needs to be addressed in IMC hardware. Can the authors provide specific information on which IMC hardware are they referring to? Can they cite studies where the occurrence of random bit flipping has been reported, outside the realm of adversarial attacks? What was the origin of such flipping and what was its prevalence?
> > >
> > >
> > > [1] Rasch, M.J. et al., "Hardware-aware training for large-scale and diverse deep learning inference workloads using in-memory computing-based accelerators" Nat Commun 14, 5282 (2023). https://doi.org/10.1038/s41467-023-40770-4

---

> > > > ### Author Response · Authors · 2024-11-26
> > > > **Answering to Reviewer 2ZTo’s Additional Questions**
> > > >
> > > > #### We thank the reviewer for the response and additional questions for clarification.
> > > >
> > > > ---
> > > > #### **[Scope of our work]**
> > > >
> > > > #### **Can the authors provide specific information on which IMC hardware are they referring to?**
> > > > #### We agree with the reviewers that IMC noise arises from hardware-specific non-idealities and can show different noise distributions. But our work is concerned about **digital IMC devices**, particularly those utilizing DRAM, RRAM, SRAM, and MRAM technologies [A, B, C, E, F], which manifests a similar granularity of bitwise errors as considered in our study. For instance, [D] demonstrated that 110 out of 129 DRAM modules exhibit disturbance errors leading to bit-flip caused by capacitive coupling and electromagnetic inference. Other studies [E, F] have shown evidence of bit flips in IMC devices, such as RRAM and SOT-MRAM, caused by process variations, thermal noise, and voltage-induced state changes.
> > > >
> > > > ---
> > > >
> > > > **Can they cite studies where the occurrence of random bit flipping has been reported, outside the realm of adversarial attacks?**
> > > > #### We are more than happy to provide prior works, outside the context of adversarial attacks, that have reported the natural (**or random**) occurrence of bit-flips in IMC devices [E, F]. Cirakoglu et al. [E] demonstrate that bit-flips in MTJ-based CRAM systems arise from faults in the magnetic states during logic operations. These bit-flips occur when voltage pulses applied to MTJ cells fail to sustain the desired resistance state, often due to variations in the magnetic or electrical properties of the device. Hoffer and Kvatinsky [F] investigate bit-flip errors in SOT-MRAM-based stateful logic gates, where process variations, thermal noise, and small margins in tunnel magnetoresistance (TMR) ratios lead to unintended state changes (bit-flips) during logic operations.
> > > >
> > > > ---
> > > >
> > > > **What was the origin of such flipping and what was its prevalence?**
> > > > #### The origin of bitwise errors in these studies is attributed to natural and operational factors within the device architectures, and they are prevalent. For instance, in the MTJ-based CRAM study [E], bit-flips are caused by voltage fluctuations during logic operations, variability in tunneling magnetoresistance (TMR) ratios, and environmental conditions such as temperature changes. These errors become more prevalent in multi-step operations as the complexity of operations increases. In the SOT-MRAM study [F], bit-flips stem from process variations, thermal noise, and small voltage margins, which lead to unintended state changes during logic operations. Monte-Carlo simulations reveal that bitwise errors are frequently caused by overlapping current values for specific input conditions or insufficient critical current in VGSOT arrays.

---

> > > > > ### Author Response · Authors · 2024-11-26
> > > > > **Rebuttal - cont'd**
> > > > >
> > > > > #### ...cont'd from the rebuttal
> > > > >
> > > > > ---
> > > > >
> > > > > #### We acknowledge that the current manuscript does not convey this scope clearly, and we will make the following updates to our manuscript to improve clarity:
> > > > >
> > > > > #### (1) We will expand our discussion to clarify that our study is concerned about **digital IMC devices**, particularly those utilizing DRAM, RRAM, SRAM, and MRAM technologies.
> > > > >
> > > > > #### (2) To address the reviewer's concern, we will expand the discussion on prior works [D, E, F], which report random bit-flips in IMC devices caused by voltage-induced state changes in MTJ-based CRAM and errors driven by process variations and thermal noise in SOT-MRAM arrays.
> > > > >
> > > > > #### (3) To further clarify the origin and prevalence of naturally occurring bit-flip errors in IMC devices, we will include additional discussion of prior works [D, E, F] in the revised manuscript.
> > > > >
> > > > >
> > > > > ---
> > > > >
> > > > > #### **[Synopsys]**
> > > > >
> > > > > #### We appreciate the reviewer’s feedback on the scope of our work.
> > > > >
> > > > > #### The current rating, however, for this work is 1, which according to the review guidelines of other premier ML conference venues such as NeurIPS, typically indicates “Very strong reject: a paper with trivial results or unaddressed ethical considerations.” We see the weaknesses in the current review do not fall into these categories. We kindly request the reviewer to reconsider the rating. Or if there is/are any remaining concerns to push this paper to “Very strong reject,” we will be happy to address them during the extended discussion phase.
> > > > >
> > > > > ---
> > > > > #### **[References Cited]**
> > > > > #### A. Jhang, Chuan-Jia, et al. "Challenges and trends of SRAM-based computing-in-memory for AI edge devices." IEEE Transactions on Circuits and Systems I: Regular Papers 68.5 (2021): 1773-1786.
> > > > > #### B. Sudarshan, Chirag, et al. "A novel DRAM-based process-in-memory architecture and its implementation for CNNs." Proceedings of the 26th Asia and South Pacific Design Automation Conference. 2021.
> > > > > #### C. Gao, Fei, Georgios Tziantzioulis, and David Wentzlaff. "Computedram: In-memory compute using off-the-shelf drams." Proceedings of the 52nd annual IEEE/ACM international symposium on microarchitecture. 2019.
> > > > > #### D. Kim, Yoongu, et al. "Flipping bits in memory without accessing them: An experimental study of DRAM disturbance errors." ACM SIGARCH Computer Architecture News 42.3 (2014): 361-372.
> > > > > #### E. Lv, Yang, et al. "Experimental demonstration of magnetic tunnel junction-based computational random-access memory." npj Unconventional Computing 1.1 (2024): 3.
> > > > > #### F. Hoffer, Barak, and Shahar Kvatinsky. "Performing stateful logic using spin-orbit torque (sot) mram." 2022 IEEE 22nd International Conference on Nanotechnology (NANO). IEEE, 2022.

---

### Official Review · Reviewer_xzTf · 2024-10-31

**Soundness:** 3
**Presentation:** 4
**Contribution:** 3
**Rating:** 5
**Confidence:** 4

**Summary:**

This paper presents a method called Hessian-Aware Training to improve the resilience of deep neural networks (DNNs) against bitwise errors in model parameters, an issue that affects DNNs deployed on error-prone, emerging computing hardware. The approach minimizes the sensitivity of models to these errors by leveraging the Hessian trace to guide training. This method is shown to reduce the proportion of sensitive parameters (those likely to cause a large accuracy drop if altered), with results indicating up to a 50% decrease in the number of high-impact bitwise errors. The paper further demonstrates how Hessian-Aware Training can support compression techniques, such as quantization and pruning, with minimal accuracy loss.

**Strengths:**

1- The proposed approach provides a novel and effective method for improving the robustness of DNNs to bitwise errors in parameters by leveraging the Hessian trace as an optimization objective during training. This strategy is unique compared to more typical hardware or system-level solutions and represents a substantial step toward integrating resilience at the model level.

2- Extensive experimental evaluation across multiple models and datasets, including MNIST, CIFAR-10, and ImageNet, illustrates the practical effectiveness of the Hessian-Aware Training method. The results strongly demonstrate improved resilience to bitwise errors and show the approach’s versatility in enhancing compression techniques, such as quantization and pruning, with minimal accuracy loss.

3- The technique has potential applications beyond model robustness, as it could complement existing hardware- and software-based resilience mechanisms. The paper discusses several examples, which underscores the broader implications of Hessian-Aware Training and suggests valuable synergies for real-world DNN deployments.

**Weaknesses:**

- The method’s reliance on the Hessian trace may limit scalability in very large models, where the computational costs of Hessian approximation could present a challenge. Although the paper addresses this with a truncated approach (focusing on top-p eigenvalues), the limitations around computational complexity remain a practical consideration.
- The results on fully connected layers appear more promising than those for convolutional layers in architectures like ResNet, where residual connections may already provide a certain level of resilience. Expanding the method’s effectiveness across different model architectures could further increase the approach’s practical impact.
- The use of an additional regularization term in training (Hessian trace) could introduce additional tuning complexity, especially in larger models. Practical guidelines on hyperparameter selection for varying model scales could be helpful for potential adopters to maximize the resilience benefits without extensive hyperparameter optimization.

**Questions:**

- Could the authors expand on potential strategies to make the method more computationally efficient, especially when dealing with very large-scale models?
- Given that convolutional layers with residual connections showed limited benefit, could the authors discuss ways to adapt the approach for models with a large proportion of residual or skip connections?
- In applications with severe hardware constraints, does the method provide enough benefit to justify the added training cost, or would alternative resilience strategies (e.g., quantization alone) be preferable?

---

> ### Author Response · Authors · 2024-11-25
>
> #### We thank the reviewer xzTf for the time and effort in evaluating our manuscript. Below, we provide answers to the reviewer’s concerns and questions. We also updated our manuscript to address the feedback. These changes are highlighted in blue.
>
> ---
>
> **[Concern 1 and question 1: overhead]**
> #### We appreciate the reviewer’s concern and agree that the reliance on the Hessian trace present scalability challenges for very large models due to the computational costs of Hessian approximation. To address this concern, we conducted additional experiments on reducing overhead. Instead of computing hessian trace across the entire parameter space, we can focus on the most sensitive layers to reduce the overhead. We applied Hessian regularization incrementally, starting with only the last layer and extending it to the last 2, 3, and finally 4 layers of the model and compared the runtime with baseline training. We run our training 5 times and report per epoch training time in Pytorch. Our results are summarized in the table below.
>
> | Model   | Dataset       | Baseline       | 1 | 2 | 3 | 4|
> |------------|----------------|----------------|----------------|----------------|----------------|----------------|
> | ResNet18 | Cifar10    |     36.24 ± 0.67 |	37.77 ± 0.39	| 43.24 ± 0.28 |	57.63 ± 0.44 |	78.24 ± 1.13 |
> | ResNet50 | ImageNet    |     7275.6 ± 18.41 |	8647.2 ± 25.43 |	10134.7 ± 30.21 |	13289.5 ± 35.76 |	16547.8 ± 42.15 |
>
> #### Our results demonstrate that training overhead increases as we increase the “layers involved in hessian-regularization.” However, using only the last 1 layer of the model, we can reduce the overhead to almost the same as baseline training, making our method efficient for very large models. We added these results in the appendix section (B.7) for enhancing the clarity on overhead of our method.
>
> ---
>
> **[Concern 2 and Question 2: Adopting our approach to large models with many skip connections]**
> #### We thank the reviewer for raising his concerns about the applicability of our method to models with a large percentage of skip connections. We respectfully want to point out that our method is compatible with models that have a large percentage of skip connections, such as visual transformers. Our experiment with the Diet-tiny model (Appendix B.4) shows that, in alignment with our results in Table 3, our method reduces erratic bits by ~7%, hence providing evidence  that our approach is compatible with models with large percentage of skip connections.
>
> ---
>
> **[Concern 3 : Providing hyperparameter selection guidelines]**
> #### We agree with the reviewer that including an additional hessian regularization term can introduce tuning complexity, particularly for larger models. We clarify that, we use a computationally inexpensive approach (section 4.3, methodology) to determine suitable values for the regularization coefficient $\lambda$ across different models. Using a small mini-batch of data, we train models 5 epochs and observe the model's sensitivity. We use the regularization coefficient $\lambda$ that minimizes sensitivity. Our experiments show that for larger models, $\lambda$ value of $10^{-2}$ works quite well, and for small MNSIT models, $\lambda$ value 1 is suitable. To address the reviewer's concern, we updated our manuscript with an additional explanation of our hyperparameter selection. These changes can be found in the section 4.3, lines 262 and 272.
>
> ---

---

> > ### Author Response · Authors · 2024-11-25
> > **Rebuttal - cont'd**
> >
> > #### ...cont'd from the rebuttal
> >
> > ---
> >
> > **[Answer to Question 3]**
> > #### Our method is orthogonal to alternative strategies, not a trade-off where we choose one or the other approach. While quantization reduces precision, Hessian-aware training does not necessarily impact the representational capacity of the model by reducing the precision. For applications that require a higher degree of precision, Hessian-aware training can thus offer an added degree of resilience. Table 5 of our paper shows that models trained with our approach retain much better accuracy than baseline training. For extreme and aggressive quantization (2-bit), the model trained with our approach retains much better accuracy than baseline models. Hence, justifies the additional cost of Hessian-aware training because it can be combined with existing quantization strategies, providing a compound effect on resilience. For clarification, we include this discussion in section 5.3 (line 465) of our revised paper.
> >
> > ---
> >
> > **[Summary]**
> > #### We thank the reviewer for the valuable feedback and detailed evaluation of our manuscript. In our rebuttal, we show that our method achieves a runtime comparable to baseline training for layer-sampled hessian regularization. We also clarify that our method is compatible with models having a large percentage of skip connections. Additionally, we extended our discussion on hyperparameter selection and showed that our method complements strategies like quantization, providing compound benefits. We kindly ask the reviewer to adjust the rating if the concerns are addressed in our rebuttal. We are open to clarifying any further concerns or answering additional questions during the discussion phase.

---

### Official Review · Reviewer_SkKx · 2024-11-04

**Soundness:** 2
**Presentation:** 2
**Contribution:** 2
**Rating:** 3
**Confidence:** 4

**Summary:**

This paper proposes using Hessian trace to enhance the model's resilience to a bit-flipping error by changing the loss landscape and avoiding model going into sharp local optima.

**Strengths:**

Strength:
* Complete experiments results from accurcay to loss landscape visualization

**Weaknesses:**

Weakness:
* The results part is vague without the mention of model accuracy under bit-wise error. For example, I am quite confused about Table 3, where the baseline even has a better accuracy. I am wondering whether the authors should report accuracy under attack and also show the trend of accuracy change with increasing error ratio.
* The error generation is not a fixed policy for all models. The authors use bitwise errors for the easy task, MINIST, while only flipping the MSB for IMAGNET. I don't think it is a fair setting for experiments.
* Moreover, using Hessian to avoid the model going into sharp loss and increase the model's resilience to bit-error is not a new idea, which is already widely explored in quantization, weakening the novelty of this paper.

**Questions:**

Please see weakness.

---

> ### Author Response · Authors · 2024-11-24
>
> #### We thank reviewer SkKx for the time and effort in reading and evaluating our manuscript. Below, we answer the reviewer’s concerns and questions about understanding our result, fairness in the experiment setup and novelty. To address the concerns, we revised our manuscript and highlighted the changes in blue.
>
> ---
>
> **Concern 1: Understanding results in Table 3**
> #### We thank the reviewer for raising concerns over the results in Table 3. We respectfully disagree with the reviewer on the point where he/she mentions, “baseline even has a better accuracy.” We show in Table 3 that the validation accuracies of models trained with our method are comparable to baseline training, and in some instances, even better (see results of ResNet18 on Cifar10).
> #### We do not mention the model accuracy under bitwise error to DNN model parameters. The systematic resilience measurement framework we use [A] alters single-bits, hence, a model with a million parameters will have 1x32(bits per parameter) = 32 Million bits flips. Instead, we show how many parameters are in the model who’s flipping leads to a more than 10% RAD, to measure the resilience of a DNN under bitwise error. We want to further clarify that there is no error ratio in our study. Also, figures 1,2 and 6 show the distribution plot of accuracy degradation after bit-flips and how these ranges are getting reduced in models trained using hessian-aware training.
>
> ---
>
> **[Concern 2 : Fairness in experiment setup]**
> #### We clarify that our experimental setup balances computational feasibility with meaningful bitwise error to the model parameters across models of varying sizes. Due to the prohibitive computational time required for complete bitwise error analysis on large models, we leverage findings from MNIST and LeNet experiments—where 99.99% of erratic bits are exponents—and follow prior studies [A] to selectively analyze exponent bits flips for larger models like ResNet18 and MSB for ResNet50. This approach ensures practical yet rigorous evaluation, as a full analysis would take ~503 days for ResNet18 (11M parameters, CIFAR-10) and ~1172 days for ResNet50 (25.6M parameters, ImageNet) with our current resources (4-8 GPUs and 60-120 CPUs). It is important to note that prior works have not performed a complete analysis of bitwise errors on large-scale models simply due to the prohibitive computational demands [A, B, C]. To address the reviewer's concern, we have incorporated these additional explanations into Section 5.1 (lines 297–302) to improve clarity on the fairness of our experiment setup.

---

> > ### Author Response · Authors · 2024-11-24
> > **Rebuttal - cont'd**
> >
> > #### ...cont'd from the rebuttal
> >
> > ---
> >
> > **[Concern 3 : Novelty]**
> > #### We appreciate the reviewer’s feedback regarding the novelty of our approach and would like to clarify that no prior work has used hessian-aware model training to enhance the model’s inherent resilience to bitwise error in parameters. Prior work uses the Hessian to measure sharpness or flatness to improve model generalization and accuracy or to support model compression  [D, E, F]. Unlike previous approaches, our objective directly targets model sensitivity to bitwise error in parameters, which are common in emerging computing environments.
> >
> > #### Furthermore, our work diverges from prior works [D, F] by employing different methodologies for hessian trace estimation and eigenvalue calculation. Additionally, our training algorithm itself is different from prior works, using the mean eigenvalue to selectively regularize the model for going into sharp loss-landscapes.
> >
> > #### Lastly, our study highlights that naturally flat layers, such as those with skip connections in residual blocks, are more resilient to bitwise errors in parameter due to their flat loss landscape. This finding identifies the most error-sensitive layers (e.g., Conv and Fully Connected) and provides a foundation for designing resilient architectures for error-prone environments and emerging devices.
> >
> > #### We believe these are sufficient novelty to distinguish our work from prior approaches and the first to train models with inherent resilience to bitwise errors in parameters.
> >
> > ---
> >
> > **[Summary]**
> >
> > #### We thank the reviewer for the feedback on our manuscript. We addressed the concerns about results in table 3, the fair setting or experiments and the novelty of our work. We kindly request the reviewer to adjust the rating if our rebuttal answers the existing issues. If there is/are any remaining concern(s), we are open to addressing them in the discussion phase.
> >
> > ---
> >
> > **[References]**
> >
> > #### A. Sanghyun Hong, Pietro Frigo, Yigitcan Kaya, Cristiano Giuffrida, and Tudor Dumitras. Terminal brain damage: Exposing the graceless degradation in deep neural networks under hardware fault attacks. In 28th USENIX Security Symposium (USENIX Security 19), pages 497–514, 2019.
> >
> > #### B. Adnan Siraj Rakin, Zhezhi He, and Deliang Fan. Bit-flip attack: Crushing neural network with progressive bit search. In Proceedings of the IEEE/CVF International Conference on Computer Vision, pages 1211–1220, 2019.
> >
> > #### C. Tanj Bennett, Stefan Saroiu, Alec Wolman, and Lucian Cojocar. Panopticon: A complete in-dram rowhammer mitigation. In Workshop on DRAM Security (DRAMSec), volume 22, page 110, 2021.
> >
> > #### D. Yang, H., Yang, X., Gong, N. Z., & Chen, Y. (2022, July). Hero: Hessian-enhanced robust optimization for unifying and improving generalization and quantization performance. In Proceedings of the 59th ACM/IEEE Design Automation Conference (pp. 25-30).
> >
> > #### E. Pierre Foret, Ariel Kleiner, Hossein Mobahi, and Behnam Neyshabur. Sharpness-aware minimization for efficiently improving generalization. In International Conference on Learning Representations, 2021.
> >
> > #### F. Z. Dong, Z. Yao, A. Gholami, M. Mahoney, and K. Keutzer. Hawq: Hessian aware quantization of neural networks with mixed-precision. In 2019 IEEE/CVF International Conference on Computer Vision (ICCV), pages 293–302, Los Alamitos, CA, USA, nov 2019. IEEE Computer Society.

---

### Official Review · Reviewer_bQFx · 2024-11-05

**Soundness:** 3
**Presentation:** 3
**Contribution:** 2
**Rating:** 3
**Confidence:** 4

**Summary:**

The paper presents a training algorithm that enhances the resilience of PIM to bitwise errors in model parameters. This method uses a second-order Hessian-based approach, minimizing the sharpness of the loss landscape to improve error. The proposed method demonstrates improved resilience to single-bit errors, aids in efficient model compression (e.g., quantization and pruning), and complements existing defensive techniques.

**Strengths:**

Strength:
1.	The proposed training method make the NN parameter space smooth and make it robust to various perturbations, e.g., bitwise flip error, quantization and pruning, etc.

**Weaknesses:**

Weaknesses:

1.	The novelty of proposed method is limited. How does it differentiate with prior work in robustness-aware optimization and hardware/software co-design methods for ReRAM PIM?, e.g., HERO: Hessian-Enhanced Robust Optimization for Unifying and Improving Generalization and Quantization Performance.

2.	Since the author mentioned hardware defense, the proposed method slightly reduce the number of sensitive weights, which can hardly defend malicious bit-flip attacks which always attacks the rest vulnerable bits.

**Questions:**

1.	The novelty of proposed method is limited. How does it differentiate with prior work in robustness-aware optimization and hardware/software co-design methods for ReRAM PIM?, e.g., HERO: Hessian-Enhanced Robust Optimization for Unifying and Improving Generalization and Quantization Performance.

2.	Since the author mentioned hardware defense, the proposed method slightly reduce the number of sensitive weights, which can hardly defend malicious bit-flip attacks which always attacks the rest vulnerable bits.

3.     What are the major machine learning contributions? Using hessian trace as a penalty to improve robustness and smoothness is extensively studied in the AI/ML community.

---

> ### Author Response · Authors · 2024-11-24
>
> #### We thank reviewer bQFx for the time and effort in reading and evaluating our manuscript. Below, we answer the reviewer’s concerns and questions one by one. The manuscript has been updated to address the reviewers' concerns, with the changes highlighted in blue. The line numbers are based on the revised manuscript.
>
> ---
>
> **[Concern 1 and Question 1: Novelty and differentiation of our work from prior work]**
> #### Thank you for the careful review and enlightening questions about the novelty of our work. After carefully reading the HERO [A] paper, we found that our work differs from this prior work in terms of perturbation bounds, hessian approximation and DNN model training methodology using hessian. These differences are listed below.
> - #### Hero and Hessian aware training uses different weight perturbations to model parameters.
>     - #### Hero uses a theoretically bounded perturbation bounded by $l_{\infty}$ norm, defined as $\parallel \delta \parallel = {\parallel W_q - W \parallel}_{\infty} \leq \bigtriangledown/2$, where $W$ and $W_q$ denote the original and quantized weights, and \bigtriangledown/2 denotes the maximum quantization of each weight for a quantization bit width of $\bigtriangledown$.
>     - #### Our proposed Hessian-aware training uses single bitwise error to model parameters, which are not theoretically bounded and can change the model parameter dramatically.
>
> - #### Hessian-approximation and calculation of regularization term is different in our work and prior work [A].
>     - #### For hessian matrix $H$’s eigenvalues $\lambda$, HERO’s regularization term is calculated as $ L_r = \sum_{i}^{} \mathrm{\lambda}_{i}^{2} = E_z \parallel H_z \parallel^2, z\sim N(0, I).$ They calculate the hessian term $H_z$ using finite difference method, given by $H_z \approx \frac{\nabla L (W + hz) - \nabla L(W)}{h}$, h is a small positive number and h is a random vector along the direction of the gradient.
>     - #### Our study utilizes Hutchinson's method to approximate the trace of the Hessian, which is a faster method [D]. For a set of random vectors $v$ drawn from the Rademacher distribution, the trace is approximated as $Tr (H) = \mathbb{E} [v^T Hv]$ = $\frac{1}{N_v} = \sum_{i = 1}^{N_v} v^{T}_{i} Hv_i$. We obtain $v^T Hv$ by computing the gradient of the loss function twice, given by $v^t Hv = v^T \cdot \frac{\partial }{\partial \theta} \left( \frac{\partial L}{\partial x} \right) \cdot v$.
> - #### Our study and prior work differ in model training methods.
>     - #### Hero applies the hessian regularization term in each training step.
>     - #### Our algorithm computes the top-p eigenvalues for each minibatch and keeps a running mean of the eigenvalues for all minibatches. We only regularize the model if the current minibatch mean of eigenvalues is greater than the mean eigenvalue observed previously across all minibatches. Lines 234-239 of our paper discuss these unique approaches (lines 15-20 in algorithm 1).
>
> #### We believe these are sufficient novelty to differentiate our work from prior work. To address reviewer's concern, we have included the prior work in section 4.1 (lines 160-163) of the revised manuscript.
>
> ---
>
> **[Concern 2 and Question 2: Defending against Bit-flip Attacks]**
>
> #### We thank the reviewer for raising this concern, and we would like to clarify that we do not propose a defense against bit-flip attack, nor do we claim anywhere in our paper that ours is a defense against bit-flip attack. In lines 108-109 of our paper, we mention that the focus of our study is not defending against malicious bit-flip attacks. Malicious bit-flip attacks are targeted attacks on the DNN model’s bitwise representation, aiming towards high accuracy degradation. Instead, we are more concerned about the naturally occurring bit-flips in emerging computing platforms. Thus we believe that the reviewer’s concern falls outside the scope of our current study.

---

> ### Author Response · Authors · 2024-11-24
> **Rebuttal - cont'd**
>
> **[...cont'd from the rebuttal]**
>
> **[Question 3: Major Machine Learning Contributions]**
>
> #### First, we clarify that there has been no prior work on improving the “model’s inherent resilience” to bitwise error in their parameters. Second-order objectives have been utilized in prior works as a measure of the flatness of the loss landscape [B]. Few other studies have used hessian to improve accuracy or generalization in quantization [A, C, D]. Our study proposes a novel hessian-aware DNN model training algorithm that utilizes the trace of the hessian and top-p% eigenvalues to train DNN models resilient to bitwise error in their parameters.
>
> #### Second, our study identifies that naturally flat layers (layers with skip connections in residual blocks) are more resilient to bitwise error in parameters due to their naturally flat loss landscape. This is a significant ML contribution because it identifies the most sensitive layers in a DNN model against bitwise error in parameters (e.g., Conv and Fully Connected), laying the groundwork for future studies, which could be particularly beneficial for designing resilient architectures for error-prone computing environments and emerging devices.
>
> #### Lastly, our method benefits extreme model compression, enabling more aggressive quantization (2-bit) and pruning with higher performance retention than traditional training. This contribution is important for resource-constrained applications and positions our work as a versatile solution that enhances parameter resilience against bitwise error and efficiency against extreme parameter compression techniques.
>
> ---
>
> **[Summary]**
> #### We thank the reviewer for raising concern about the novelty and significant contribution of our paper to the ML field. In our rebuttal, we clarified our novelty and established the major ML contribution of our work. We also clarify that our work does not propose a defense against targeted bit-flip attacks. We therefore kindly request the reviewer to adjust the rating if our answer addresses the concerns and answers the questions. If there are any remaining concerns/questions, we are happy to address them during the discussion phase.
>
> ---
>
> **[References]**
>
> #### A.  Yang, H., Yang, X., Gong, N. Z., & Chen, Y. (2022, July). Hero: Hessian-enhanced robust optimization for unifying and improving generalization and quantization performance. In Proceedings of the 59th ACM/IEEE Design Automation Conference (pp. 25-30).
>
> #### B.  Yiding Jiang, Behnam Neyshabur, Hossein Mobahi, Dilip Krishnan, and Samy Bengio. Fantastic generalization measures and where to find them. In International Conference on Learning Representations, 2020.
>
> #### C.  Pierre Foret, Ariel Kleiner, Hossein Mobahi, and Behnam Neyshabur. Sharpness-aware minimization for efficiently improving generalization. In International Conference on Learning Representations, 2021.
>
> #### D.  Zhen Dong, Zhewei Yao, Daiyaan Arfeen, Amir Gholami, Michael W Mahoney, and Kurt Keutzer. Hawq-v2: Hessian aware trace-weighted quantization of neural networks. In H. Larochelle, M. Ranzato, R. Hadsell, M.F. Balcan, and H. Lin, editors, Advances in Neural Information Processing Systems, volume 33, pages 18518–18529. Curran Associates, Inc., 2020.

---

> > ### Comment · Reviewer_bQFx · 2024-11-25
> > **Thanks for the responses**
> >
> > I appreciate the authors' responses.
> > Hutchinson's method to approximate Hessian trace is also widely used, and it is not a novel method. This sampling is actually more costly than the paper I cited.
> > It is not convincing why IMC noise robustness issue contains random bitflip, which is not physically validated.
> > I remain the same score.

---

> > > ### Author Response · Authors · 2024-11-26
> > > **Answering to Reviewer bQFx’s Additional Questions**
> > >
> > > #### We thank the reviewer for the response and additional questions for clarification.
> > >
> > > ---
> > >
> > > #### **Hutchinson's method to approximate Hessian trace is also widely used, and it is not a novel method.**
> > >
> > > #### We clarify that approximating the Hessian using Hutchinson’s method is **not** the main novelty we claim in our paper. The core contribution lies in our approach, which demonstrates that models **trained** to reduce the Hessian can achieve resilience to parameter-level corruptions. This resilience ultimately makes models stable when they are deployed on devices prone to bitwise errors (that may or may not be adversarially exploited).
> > >
> > > ---
> > > #### **This sampling is actually more costly than the paper I cited.**
> > >
> > > #### We clarify this is **not** the case. To validate, we compare the wall time required to train a model for one epoch between our approach and the prior work [A] mentioned by the reviewer. We run this comparison with the ResNet50 model on ImageNet. The table below shows our results:
> > >
> > > | Method | Model | Dataset | Training Time |
> > > | -------- | ------- |------- |------- |
> > > | HERO [A] | ResNet50 |ImageNet | 9329.43 |
> > > | Ours | | |8647.20 |
> > >
> > > #### We observe that HERO [A] takes more time to train. Given that training with the Hessian is less stable than standard model training, HERO, without the techniques we develop such as min-max optimization, will take more training iterations to converge to a well-trained model.
> > >
> > > ---
> > > #### **It is not convincing why IMC noise robustness issue contains random bitflip, which is not physically validated.**
> > > #### We first clarify that our study focuses on **digital IMC devices**, not analog ones. We also want to point out that digital IMC devices inherently exhibit single-bit flips due to their reliance on DRAM, SRAM or MRAM architectures [B, C, D].
> > >
> > > #### We believe that there are two approaches to studying model resilience to bitwise errors in digital IMC devices: (1) Monte-Carlo simulation that randomly causes multiple numbers of bitwise errors in the memory representation of a model and measures accuracy loss. (2) Measuring accuracy degradation under an atomic error (a single bit-flip) across all the bits in a model.
> > > #### Prior work [E] using the approach (1) shows an accuracy drop of 5-10%. However, these results reflect the average cases, rather than the worst-possible scenarios (an error that can degrade accuracy by more than 10%). To consider both average and the worst-case accuracy loss under bitwise corruptions, we systematically flip every bit and evaluate their impact on model performance before and after a model trained with our Hessian-aware approach.
> > >
> > > ---
> > > **[References Cited]**
> > >
> > > #### A. Gao, Fei, Georgios Tziantzioulis, and David Wentzlaff. "Computedram: In-memory compute using off-the-shelf drams." Proceedings of the 52nd annual IEEE/ACM international symposium on microarchitecture. 2019.
> > >
> > > #### B. Kim, Yoongu, et al. "Flipping bits in memory without accessing them: An experimental study of DRAM disturbance errors." ACM SIGARCH Computer Architecture News 42.3 (2014): 361-372.
> > >
> > > #### C. Lv, Yang, et al. "Experimental demonstration of magnetic tunnel junction-based computational random-access memory." npj Unconventional Computing 1.1 (2024): 3.
> > >
> > > #### D. Hoffer, Barak, and Shahar Kvatinsky. "Performing stateful logic using spin-orbit torque (sot) mram." 2022 IEEE 22nd International Conference on Nanotechnology (NANO). IEEE, 2022.
> > >
> > > #### E. Yao, Fan, Adnan Siraj Rakin, and Deliang Fan. "{DeepHammer}: Depleting the intelligence of deep neural networks through targeted chain of bit flips." 29th USENIX Security Symposium (USENIX Security 20). 2020.

---

### Meta-Review · Area_Chair_pbQG · 2024-12-16

**Metareview:**

This paper proposes a Hessian-aware training approach to smooth the optimization loss landscape and improve the model’s robustness to single-bit flip errors. The proposed method can also help improve other model compression methods, such as quantization and pruning.

Main weaknesses:
- Limited novelty due to so many Hessian-aware approaches
- Relatively limited evaluation due to computational cost
- Hessian trace calculation may be limited to smaller models
- As mentioned by several reviewers, the noise studied in this paper is not necessarily for in-memory computing, and the evaluation across different datasets could be more consistent

**Additional Comments On Reviewer Discussion:**

During the rebuttal, the authors provided additional experiments, such as training overhead and comparison to HERO. However, I don’t think the main weaknesses are addressed sufficiently.

---

### Decision · Program_Chairs · 2025-01-22

Reject